# Doxycycline Inhibition of a Pseudotyped Virus Transduction Does Not Translate to Inhibition of SARS-CoV-2 Infectivity

**DOI:** 10.3390/v13091745

**Published:** 2021-09-01

**Authors:** Luisa Diomede, Sara Baroni, Ada De Luigi, Arianna Piotti, Jacopo Lucchetti, Claudia Fracasso, Luca Russo, Valerio Bonaldo, Nicolò Panini, Federica Filippini, Fabio Fiordaliso, Alessandro Corbelli, Marten Beeg, Massimo Pizzato, Francesca Caccuri, Marco Gobbi, Emiliano Biasini, Arnaldo Caruso, Mario Salmona

**Affiliations:** 1Department of Molecular Biochemistry and Pharmacology, Istituto di Ricerche Farmacologiche Mario Negri IRCCS, 20156 Milano, Italy; sara.baroni@marionegri.it (S.B.); ada.deluigi@marionegri.it (A.D.L.); arianna.piotti@marionegri.it (A.P.); jacopo.lucchetti@marionegri.it (J.L.); claudia.fracasso@marionegri.it (C.F.); luca.russo@marionegri.it (L.R.); fabio.fiordaliso@marionegri.it (F.F.); alessandro.corbelli@marionegri.it (A.C.); marten.beeg@marionegri.it (M.B.); marco.gobbi@marionegri.it (M.G.); 2Department of Cellular, Computational and Integrative Biology (CIBIO), University of Trento, 38122 Trento, Italy; valerio.bonaldo@unitn.it (V.B.); massimo.pizzato@unitn.it (M.P.); emiliano.biasini@unitn.it (E.B.); 3Dulbecco Telethon Institute, University of Trento, 38122 Trento, Italy; 4Department of Oncology, Istituto di Ricerche Farmacologiche Mario Negri IRCCS, 20156 Milano, Italy; nicolo.panini@marionegri.it; 5Section of Microbiology, Department of Molecular and Translational Medicine, University of Brescia, 25121 Brescia, Italy; federica.filippini@unibs.it (F.F.); francesca.caccuri@unibs.it (F.C.); arnaldo.caruso@unibs.it (A.C.)

**Keywords:** SARS-CoV-2, COVID-19, spike protein, tetracyclines, doxycycline, in vitro, surface plasmon resonance

## Abstract

The rapid spread of the pandemic caused by the SARS-CoV-2 virus has created an unusual situation, with rapid searches for compounds to interfere with the biological processes exploited by the virus. Doxycycline, with its pleiotropic effects, including anti-viral activity, has been proposed as a therapeutic candidate for COVID-19 and about twenty clinical trials have started since the beginning of the pandemic. To gain information on the activity of doxycycline against SARS-CoV-2 infection and clarify some of the conflicting clinical data published, we designed in vitro binding tests and infection studies with a pseudotyped virus expressing the spike protein, as well as a clinically isolated SARS-CoV-2 strain. Doxycycline inhibited the transduction of the pseudotyped virus in Vero E6 and HEK-293 T cells stably expressing human receptor angiotensin-converting enzyme 2 but did not affect the entry and replication of SARS-CoV-2. Although this conclusion is apparently disappointing, it is paradigmatic of an experimental approach aimed at developing an integrated multidisciplinary platform which can shed light on the mechanisms of action of potential anti-COVID-19 compounds. To avoid wasting precious time and resources, we believe very stringent experimental criteria are needed in the preclinical phase, including infectivity studies with clinically isolated SARS-CoV-2, before moving on to (futile) clinical trials.

## 1. Introduction

Since the pandemic of coronavirus disease 2019 (COVID-19) broke out in December 2019 [1], the scientific community and drug companies have been searching for compounds to interfere with the biological processes exploited by the severe acute respiratory syndrome coronavirus 2 (SARS-CoV-2) to infect cells and spread, so as to fight the pandemic.

Vaccines against the spike (S) viral protein, responsible for the virus attachment and entry into target cells, have been developed in record time. Although effective against wild-type SARS-CoV-2, their use is still limited, confined to the richest countries, leaving many world areas excluded, particularly the poorest and more populated areas, allowing the virus to spread and mutate. These vaccines, therefore, partially lose their effectiveness against emerging variants, and vaccination alone may not be enough to stop the pandemic, so host- and virus-targeted pharmacological therapy is urgently needed. 

SARS-CoV-2 enters the cells by binding the S structural protein, particularly the S1 subunit containing the receptor-binding domain (RBD), to the host cell surface receptor angiotensin-converting enzyme 2 (ACE2) [2]. Once the S protein is cleaved by the host transmembrane serine protease 2 (TMPRSS2) at the S1/S2 junction, the virus is endocytosed by the cell [2]. With the contribution of several enzymes, particularly 3-chymotrypsin-like cysteine protease (3CLpro or M^Pro^), viral genomic RNA starts to replicate and is incorporated into newly produced viral particles. The virions formed are then transported to the cell surface and released by exocytosis into the extracellular space. All these processes offer potentially druggable targets to affect virus entry, proteolysis, replication, assembly, and/or release [3]. 

Great efforts have been made, and still are, to design new drugs for treating COVID-19, although a ‘final’ therapy is unlikely to be rapidly developed and clinically approved, even in this emergency scenario. Another potential strategy is drug-repurposing, searching for an effective molecule among those already existing and approved.

In this context, the tetracycline antibiotics have been proposed as candidates against SARS-CoV-2. 

With their pleiotropic features, including anti-inflammatory and antioxidant properties, and their ability to chelate zinc compounds on matrix metalloproteinases [4], tetracyclines have been proposed, alone or combined with other compounds, as potential therapeutic candidates against COVID-19 [5,6,7,8,9,10,11]. Second-generation tetracyclines, such as minocycline and doxycycline, exerted a direct antiviral effect and can inhibit the replication of different viruses, including the human immunodeficiency virus (HIV), West Nile virus, Japanese encephalitis virus, and influenza virus [12]. In addition, tetracyclines can pass the blood–brain barrier [13], protecting central nervous system cells from the harmful effects of viral infection [14].

In silico docking studies suggested a direct interaction of tetracyclines with the RBD, which can result in inhibition of the RBD–ACE2 complex [15] and reported their ability to inhibit 3CLpro [16], thus interfering with virus internalization and replication. This antiviral activity was confirmed in a study on Vero E6 cells infected with a clinically isolated SARS-CoV-2 strain (IHUMI-3) showing that doxycycline, at concentrations compatible with the circulating levels reached after oral or intravenous administration, inhibited virus entry and replication [17]. 

Tetracyclines, alone or together with colchicine, have therefore been given to COVID-19 patients in a non-hospital setting and have been reported to improve symptoms and hasten recovery in case reports [18] and observational clinical studies [19,20]. Doxycycline combined with Ivermectin has recently resulted in better symptomatic relief, shortened recovery duration, fewer adverse effects, and superior patient compliance compared to the Hydroxychloroquine-Azithromycin combination in patients with mild to moderate COVID-19 [11]. 

About twenty clinical trials have started on tetracyclines and COVID-19 since the beginning of the pandemic [21]. The UK Platform randomized trial of interventions against COVID-19 in older people (PRINCIPLE) investigates the effect of doxycycline administered at home in the early stages of COVID-19 to patients aged over 50 [22]. The study was stopped for futility in March 2021 because the interim analysis indicated only a small benefit in terms of the recovery of symptoms and hospitalization rates in participants receiving doxycycline [22]. 

It cannot be excluded that these negative results reflect the small amount of work on the in vitro characterization of the mechanisms underlying the possible effect of tetracyclines on SARS-CoV-2. 

With this in mind, we designed experiments aimed at gaining information on the antiviral activity of doxycycline, using an integrated platform we developed to identify molecules active against SARS-CoV-2. This platform comprises in vitro binding tests and infection studies with a pseudotyped virus expressing the S protein as well as a clinically isolated SARS-CoV-2 strain. Doxycycline effectively inhibited the transduction of the pseudotyped virus but did not affect the entry and replication of SARS-CoV-2. Even though this result is disappointing, we hope this negative experience will help define more stringent categories of judgment to improve the initial selection of potentially active molecules. 

## 2. Materials and Methods

### 2.1. Cells

Human embryonic kidney (HEK) 293-T and African green monkey kidney Vero E6 cells were obtained from the American Type Culture Collection (ATCC). HEK293-T cells stably expressing human receptor ACE2 (HEK293-ACE2) were generated as described [23]. All cell lines were maintained in Dulbecco modified Eagle Medium (DMEM; Gibco/Euroclone #ECB7501L) containing 10% heat-inactivated fetal bovine serum (FBS, Gibco #10270), l-glutamine (Gibco, #25030-024), non-essential amino acids (Gibco/Euroclone, #ECB3054D), and penicillin/streptomycin (Corning, #20-002-Cl). HEK293-ACE2 required puromycin (Genespin, Milano, Italy). Cells were cultured in 100 mm^2^ Petri dishes or T75 flasks at 37 °C in a humidified 5% CO_2_ and routinely split every 3–4 days. Cells used in this study had not been passaged more than 20 times from the original stock.

### 2.2. Generation of Pseudotyped Virus Particles

Retroviral particles exposing the SARS-CoV-2 S protein were produced as described by Massignan et al. [23]. Briefly, HEK293-T cells were seeded into 10 cm plates with DMEM containing 0.5 mg/mL geneticin G418 (Thermofisher, Waltham, MA, USA). Once the cells reached approximately 80% confluence, the medium was replaced with DMEM containing 2.5% FBS. Cells were then transfected with a combination of the following plasmids: pc Gag-pol MLV packaging plasmid, pc Spike ΔC ENV-encoding vector containing the SARS-CoV-2 S as surface glycoprotein, and pc NCG MLV transfer vector containing eGFP [23]. Control retroviral particles were obtained by transfecting the cells only with the packaging and transfer vectors, missing out the plasmid encoding for SARS-CoV-2 S (No-Spike). Supernatants were collected and centrifuged at 2000× *g* for 5 min, then filtered using a 0.45 μm filter, and ultracentrifuged at 20,000× *g* for 2 h. Pellets were resuspended in 5 mM phosphate buffered saline (PBS), pH 7.4, and stored at −80 °C until use.

### 2.3. Transmission Electron Microscopy

A suspension of retroviral particles exposing the SARS-CoV-2 S protein was gently resuspended in 10 µL of 5 mM PBS and deposited on copper grids for 20 min. After absorbing the excess of the suspension with Whatman filter paper, the grids were fixed for 30 min with 0.12 M phosphate buffer solution containing 2% glutaraldehyde and 4% paraformaldehyde, rinsed in distilled water, and negatively stained with 0.1% uranyl acetate. Images were then obtained with an energy filter transmission electron microscope (EFTEM, ZEISS LIBRA^®^ 120, Carl Zeiss S.p.A., Milano, Italy) coupled with an yttrium aluminum garnet (YAG) scintillator slow-scan CCD camera (Sharp eye, TRS, Milano, Italy).

### 2.4. Transduction Assay

Different experimental settings were used for HEK293-ACE2 and Vero E6 cells. HEK293-ACE2 cells were seeded on 96-well plates (2 × 10^4^ cells/well) in complete DMEM medium. After 24 h incubation at 37 °C in humidified 5% CO_2_, the medium was replaced with fresh medium containing 0.1–100 µM doxycycline hyclate (Fagron, Quarto Inferiore, Bologna, Italy) or gentamicin sulfate (Caelo, Hilden, Germany) dissolved in Milli-Q water. Control cells were treated with equivalent volumes of water (vehicle). Cells were incubated for 4 h at 37 °C in humidified 5% CO_2_, then infected with 3 µL of retroviral vector exposing the SARS-CoV-2 S protein, or control. The day after the transduction, the medium was replaced with fresh medium, and after 24 h incubation, the transfection efficiency was checked by determining the percentage of cells expressing GFP, using an EnSight multimode microplate reader (Perkin Elmer, Milano, Italy) and a ZOE^TM^ fluorescent cell imager (Bio-Rad, Hercules, CA, USA). The ZOE^TM^ images were analyzed with the Fiji software, an open-source platform for biological-image analysis (see Appendix A). 

Untransfected Vero E6 cells were seeded on 96-well plates (7.5 × 10^3^ cells/well) in complete DMEM medium. After 24 h at 37 °C in humidified 5% CO_2_, the medium was replaced with fresh medium containing the compound to be tested at the desired concentration [23]. Control cells were treated with equivalent volumes of water (vehicle). To increase the number of transduced cells, on days 3 and 4, 3 μL of the vector exposing the SARS-CoV-2 S protein or control was added to each well. Three days after the incubation, the transduction efficiency was determined as described above. 

### 2.5. Cell Viability

Cells were seeded on 96-well plates (7.5 × 10^3^ Vero cells/well and 2 × 10^4^ HEK293-ACE2 cells/well) in complete DMEM medium with 10% FBS. After 24 h at 37 °C in humidified 5% CO_2_, the medium was replaced with fresh medium containing 0.1–100 µM doxycycline hyclate or gentamicin sulfate dissolved in Milli-Q water. Control cells were treated with equivalent volumes of water (vehicle). Cells were incubated for 24 h at 37 °C in humidified 5% CO_2_, then the medium was replaced with fresh medium and cells were incubated for another 24 h (HEK293-ACE2) or 48 h (Vero E6). Cells were then treated for 15 min up to 4 h at 37 °C with 5 mg/mL 3-(4,5-dimethylthiazol-2-yl)-2,5-diphenyltetrazolium bromide (MTT) (Sigma Aldrich, St. Louis, MO, USA, #M5655-1G) in 5 mM PBS. The MTT was carefully removed, and cells were resuspended in acidified isopropanol (0.04 M HCl) or 60 µL DMSO; cell viability was determined by measuring the absorbance at 560 nm using a spectrophotometer (Infinite M200, Tecan, Männedorf, Switzerland).

### 2.6. Cell Cycle

Monoparametric analysis of DNA was performed on exponentially growing HEK293-ACE2 cells. Cells were seeded on 12-well plates (2.4 × 10^5^ cells/well) in complete DMEM medium with 10% FBS. After 24 h at 37 °C in humidified 5% CO_2_, the medium was replaced with fresh medium containing 1 or 100 µM doxycycline hyclate in Milli-Q water. Control cells were treated with equivalent volumes of water (vehicle). The cell cycle perturbation was evaluated before and 6, 24, 30, and 48 h after the treatment. Cells were counted using a Vi-CELLTM XR cell viability analyzer (Beckman Coulter, Brea, CA, USA) and fixed at 4 °C in 70% ethanol for at least 24 h before staining. For this, 2 × 10^6^ cells were incubated overnight at 4 °C with 1 mL of a solution containing 25 µg propidium iodide and 12.5 µL RNAse. DNA flow cytometric analyses were performed on at least 1 × 10^4^ cells at the acquisition rate of 300 events per second, using a Gallios flow cytometer (Beckman Coulter). Doublets were excluded from the analyses.

### 2.7. Western Blot Analysis

HEK293-ACE2 cells were seeded on 12-well plates (2.4 × 10^5^ cells/well) in complete DMEM medium with 10% FBS and incubated for 24 h at 37 °C in humidified 5% CO_2_. The medium was replaced with fresh medium containing 0.1–100 µM doxycycline hyclate in Milli-Q water. Control cells were treated with equivalent volumes of water (vehicle). After 3 and 6 h, the medium was removed, cells were collected and lysed for 15 min at 4 °C with 20 mM Tris-HCl solution, pH 7.5, containing 150 mM NaCl, 1 mM Na_2_EDTA, 1 mM EGTA, 1% NP-40, 1% sodium deoxycholate, 2.5 mM sodium pyrophosphate, 1 mM β-glycerophosphate, 1 mM Na_3_VO_4_, 1 µg/mL leupeptin. Samples were centrifuged for 10 min at 16,100× *g* and the protein content in the lysates was quantified with a BCA protein assay kit (Thermofisher). Samples were then immunoblotted using 10% bis-Tris gel (Invitrogen, Waltham, MA, USA) and transferred to a PVDF membrane (Millipore, Vimodrone, Milano, Italy); 25 µg of total proteins were loaded in each lane of the gel. The membranes were incubated overnight at 4 °C with anti-ACE2 mouse monoclonal antibody AC18Z (1:2000, Santa Cruz Biotechnology, Santa Cruz, CA, USA) or anti-β-actin mouse monoclonal antibody (1:5000, Sigma Aldrich). Anti-mouse IgG peroxidase conjugated (1:5000, Sigma Aldrich) was used as secondary antibody. Hybridization signals were detected with a ChemiDoc XRS Touch Imaging System (Bio-Rad). 

### 2.8. Doxycycline Stability and Binding to Albumin

Doxycycline was incubated at 1, 10, and 100 µM in 500 µL of DMEM medium added to HEK293-ACE2 cells seeded on 96-well plates (as in the transduction assay, see above). Stability and the bovine serum albumin (BSA)-bound fraction of doxycycline was assessed after 5 min and 0.5, 1, 2, 4, 6, and 24 h of incubation at 37 °C in humidified 5% CO_2_ (each well, in duplicate, corresponded to a different incubation time). At each timepoint, medium was removed, and an aliquot was used for determination of the doxycycline concentration. 

Doxycycline binding to BSA was assessed only for 10 µM concentration. BSA-bound and free doxycycline were separated by ultrafiltration using Amicon Ultra-0.5 centrifugal filter devices (Merck Millipore, Vimodrone, Milano, Italy) with a MW cutoff of 30 KDa. Doxycycline in the three fractions (total, unbound, and BSA-bound) was measured using a validated HPLC-MS/MS method [13]. The amounts in the unbound and BSA-bound fractions were calculated using a mass balance approach to minimize inaccuracy due to confounding factors (e.g., non-specific binding of doxycycline to the filter membrane) [24].

### 2.9. Surface Plasmon Resonance

All analyses were performed with a ProteOn XPR36 protein interaction array system (Bio-Rad Laboratories, Hercules, CA) surface plasmon resonance (SPR) apparatus with six parallel flow channels that can immobilize up to six ligands on the same sensor chip. FLAG-tagged ACE2 (AdipoGen, Liestal, Switzerland) was captured on the chip by a previously immobilized anti-FLAG antibody (Merck Life Science S.r.l, Milano, Italy). S protein (Euprotein, North Brunswick, NJ, USA), its S1 domain and its RBD (SinoBiological, Wayne, PA, USA), all Fc-tagged, were captured on the same chip by a previously immobilized anti-Fc antibody (Merck Life Science). Anti-FLAG or anti-Fc antibodies were immobilized by classical amine coupling chemistry [25] flowing them for 5 min at 30 µg/mL in acetate buffer, pH 5.0, on GLC sensor chips pre-activated as described by the producer (Bio-Rad); the remaining activated groups were blocked with ethanolamine (pH 8.0.), FlagACE-2, FcS, FcRBD, or FcS1 were then flowed on the corresponding anti-tag antibodies at 30 µg/mL in 10 mM phosphate buffer containing 150 mM NaCl and 0.005% Tween 20 (PBST, pH 7.4), also used as running buffer. Two flow channels were prepared in parallel with the two capturing antibodies only, as reference surfaces. The level of immobilization ranged from 1000 to 2200 resonance units (RU) (Appendix A). The flow channels were rotated 90° so that up to six analyte solutions could be flowed in parallel on all the immobilized ligands, creating a multi-spot interaction array. 

To evaluate the direct binding of doxycycline on all the proteins captured simultaneously, we used the “kinetic titration” design [26]. The drug was injected at concentrations from 1 to 100 µM, in PBST, pH 7.4, with short dissociation times in between, with no regeneration steps. To evaluate the ability of doxycycline to inhibit the ACE2-RBD interaction, we preincubated 10 nM ACE2 (or 60 nM RBD) for 60 min at room temperature with or without the drug, and then injected the mixture over chip-immobilized RBD (or ACE2). All SPR assays were run at a rate of 30 µL/min at 25 °C. The sensorgrams (time course of the SPR signal in RU) were normalized to a baseline of 0.

### 2.10. Virus

We successfully isolated SARS-CoV-2 in Vero E6 cells from the nasopharyngeal swab of a COVID-19 patient [27]. The identity of the strain was verified by metagenomic sequencing, from which the reads mapped to nCoV-2019 (genomic data are available at EBI under study accession n° PRJEB38101). The clinical isolate was propagated in Vero E6 cells, and the viral titer was determined by a standard plaque assay. Infection experiments were conducted in a biosafety level-3 laboratory (BLS-3) at a multiplicity of infection (MOI) of 0.01.

### 2.11. Authentic Virus Infection Assay

Vero E6 cells were treated for 4 h with 100 µM doxycycline or gentamicin, then were infected, for 1 h, in the presence of 100 µM doxycycline or gentamicin with the SARS-CoV-2 isolate at a MOI of 0.01. Infection was performed in DMEM medium without FBS. Then, after the removal of the virus and washing with warm PBS, cells were cultured in a medium containing 2% FBS with 100 µM doxycycline or gentamicin. As a control, Vero E6 cells were infected with or neither antibiotic. At 48 h post infection, cells and supernatants were collected for further viral genome quantification.

### 2.12. Viral RNA Extraction and qRT-PCR

RNA was extracted from clarified cell culture supernatants (16,000× *g* for 10 min) and infected cells using a QIAamp Viral RNA MiniKit and RNeasy Plus mini kit (Qiagen, Hilden, Germany), respectively. RNA was eluted in 30 μL of RNase-free water and stored at –80 °C until use. qRT-PCR was carried out as previously described [28]. Briefly, reverse transcription and amplification of the S gene were performed with the one-step QuantiFast Sybr Green RT-PCR mix (Qiagen) as follows: 50 °C for 10 min, 95 °C for 5 min, 95 °C for 10 s, 60 °C for 30 s (40 cycles) (primers: RBD-qF1: 5′-CAA TGG TTT AAC AGG CAC AGG-3′ and RBD-qR1: 5′-CTC AAG TGT CTG TGG ATC ACG-3). A standard curve was obtained by cloning the RBD of S gene (primers: RBD-F: 5′-GCT GGA TCC CCT AAT ATT ACA AAC TTG TGCC-3′; RBD-R: 5′-TGC CTC GAG CTC AAG TGT CTG TGG ATC AC-3′) into pGEM T-easy vector (Promega, Madison, WI, USA). A standard curve was generated by determining the copy numbers derived from serial dilutions of the plasmid (10^3^–10^9^ copies). Each quantification was run in triplicate.

### 2.13. Statistical Analysis

For statistical analyses, Prism GraphPad software v.7.03/8.0 (GraphPad Software, San Diego, CA, USA) was used, including all the data points, with the exception of experiments in which negative and/or positive controls did not give the expected outcome. No test for outliers was employed. The results were expressed as means ± SD or SEM. The data were analyzed with one-way ANOVA, including analysis of the normality of data, and corrected by a Bonferroni or Dunnett post hoc test. Probability at *p* ˂ 0.05 was considered significant; 50% inhibitory concentration (IC_50_) values were obtained by fitting dose-response curves to three-parameter non-linear fit (to a sigmoidal function using a 4PL non-linear regression model).

## 3. Results

### 3.1. Effect of Doxycycline on the Transduction of a Pseudotyped Retroviral Vector Exposing the SARS-CoV-2 S Protein

We first investigated doxycycline’s ability to counteract SARS-CoV-2 infection using a pseudotyped retroviral vector exposing the SARS-CoV-2 S protein and expressing a GFP reporter gene [23] (Figure 1A). This retrovirus was spherical with a diameter of 140–160 nm, surrounded by a lipid bilayer envelope. Spikes with length from 15 to 27 nm were embedded in the envelope and penetration of the negative stain into the retrovirus revealed the viral capsid.

Vero E6 and HEK293-ACE2 cells, both expressing the ACE2 receptor, were incubated with different concentrations of doxycycline and transduced with retroviral vectors pseudotyped with the SARS-CoV-2 S protein, or control vectors (No-Spike). Cells were treated in the same experimental conditions with gentamicin, an antibiotic structurally related to doxycycline but without of the pleiotropic activity of tetracyclines [4]. We estimated the effect of each compound on retroviral vector transduction by quantifying the percentages of cells presenting GFP fluorescence. Blinded analysis indicated that doxycycline inhibited the retroviral transduction in Vero E6 (Figure 1B) and HEK293-ACE2 (Figure 1C) cells in a dose-dependent manner. No transduction was observed when Vero E6 cells (Figure 1D) or HEK293-ACE2 (data not shown) were infected with the No-Spike control vector. Gentamicin did not significantly modify GFP transduction in either cell line indicating the specificity of doxycycline’s effect (Figure 1E,F and Appendix A).

Doxycycline was more effective in inhibiting the transduction of Vero E6 than HEK293-ACE cells, as indicated by the IC_50_ (16.92 ± 1.55 µM for Vero E6 and 58.81 ± 1.45 µM for HEK293-ACE2, *p* < 0.0001) or comparing the effects of 100 µM doxycycline on the two cell types; we observed 62.5 % and 38.5% inhibition of transduction in Vero E6 and HEK293-ACE2, respectively (Figure 1D). 

No degradation of doxycycline occurred during the 24 h incubation in cell medium (Appendix A). We also found that about 80% of doxycycline was bound to BSA in the medium (Appendix A), confirming its marked ability (80–90%) to bind to plasma proteins [29].

The difference in GFP transduction efficiency between the two cell lines cannot be ascribed to a toxic effect of doxycycline, which did not induce significant cytotoxicity in Vero E6 and HEK293-ACE2 cells (Appendix A) nor affected the proliferation of HEK293-ACE2 cells (Appendix A). Since gene transfer by retroviral vectors can occur only in cells that are actively replicating at the time of infection, we also investigated whether doxycycline affected the cell cycle of HEK293-ACE2. There was no change in the DNA content in the different phases of cell cycle in cells treated with 1 or 100 µM doxycycline at all time points (Appendix A). In addition, doxycycline did not affect the level of ACE2 expression in HEK293-ACE2 cells (Appendix A). 

These results indicate that doxycycline may reduce cellular entry for a pseudotyped retroviral vector exposing the SARS-CoV-2 S protein and that efficacy may be related to the cell type. 

### 3.2. SPR Studies

SPR studies were conducted to determine whether doxycycline reduces retroviral transduction by binding to the S protein and/or ACE2. No evidence of a doxycycline binding, up to 100 μM, to ACE2, S, S1, nor RBD was obtained in SPR studies using a direct approach (i.e., flowing the drug over immobilized proteins) (Appendix A). However, the possibility of false negative data cannot be excluded, as SPR has lower sensitivity of SPR when testing small molecules. For this reason, we also used a different SPR approach to see whether doxycycline inhibited the RBD-ACE2 interaction. This can be detected well by SPR, either flowing ACE2 (10 nM) over immobilized RBD (Figure 2A, purple line, estimated Kd = 0.9 nM) or, vice versa, flowing RBD (60 nM) over immobilized ACE2 (Figure 2B, purple line, estimated Kd = 1.4 nM). Preincubation of ACE2 or RBD with 100 μM doxycycline for 60 min at room temperature, in solution (Figure 2A,B, red lines), did not affect the binding of the protein with the partner immobilized on the sensor chip (RBD or ACE2, respectively), suggesting that the drug did not occupy the relevant binding sites to a significant extent.

### 3.3. Effect of Doxycycline on Authentic SARS-CoV-2 Strain Replication

To determine the ability of doxycycline to counteract the infectivity of SARS-CoV-2, Vero E6 cells were pretreated for 4 h with 100 µM doxycycline or gentamicin before the infection with the authentic SARS-CoV-2 strain at a MOI of 0.01. Cells were then washed and cultured for 48 h in fresh medium containing 100 µM doxycycline or gentamicin. As shown in Figure 3A, SARS-CoV-2 induced cytolytic effects on Vero E6 cells which was not modified by the treatment with doxycycline or gentamicin. Quantification of viral RNA copy number in the cell culture supernatants (Figure 3B) and at intracellular levels (Figure 3C) indicated that doxycycline did not exert any inhibitory effects on viral particles production and genome expression, respectively. These findings indicated that doxycycline, although effective in the pseudotyped virus transduction assay, did not inhibit SARS-CoV-2 replication.

## 4. Discussion 

The rapid spread of the pandemic caused by the SARS-CoV-2 virus has created an unusual situation in defining the strategies to develop vaccines or antiviral drugs in a broad sense. The pandemic surprised everyone by the speed of its spread and, above all, by the absence of integrated national and international defense strategies [3]. 

The development of medicines usually takes a very long time between conception and the availability to the patient. Still, in the COVID-19 case, the time factor was decisive. Therefore, the scientists aimed at developing vaccines and antiviral medicines, reducing the time for their availability as much as possible. Of course, this new scenario has substantially changed the timing of drug development which has also resulted in the generation of many false-negative or false-positive results [3]. 

The possibility of using artificial intelligence to identify potential molecules active against the spread of the pandemic has prompted many groups to carry out in silico studies and screen entire libraries [30,31]. 

In the case of anti-COVID-19 drugs, numerous molecules have been identified through in silico studies as potentially active, but in reality, the outcome of this kind of approach has not been as successful as expected. Many of the molecules identified in silico have reported controversial results proving that the transition from in silico screening to the clinical application requires great caution and careful studies to verify the in vitro efficacy. It is, therefore, necessary to establish new paradigms for evaluating the efficacy of a potentially active molecule.

As an example, in this paper, we report the controversial results obtained with doxycycline, which in some way echo those already published in the literature. We demonstrated for the first time that doxycycline significantly inhibited the transduction of a pseudotyped virus on two different cell lines. However, this effect did not translate into the drug’s ability to counteract in vitro in Vero E6 cells the entry and replication of the authentic SARS-CoV-2 virus. This finding was in contrast with that previously reported by Gendrot and collaborators [17] which, using Vero E6 cells too, found doxycycline effective in counteracting SARS-CoV-2 infectivity. It cannot be excluded that this discrepancy could be due to the different SARS-CoV-2 strains used to infect the cells. We used a clinically isolated SARS-CoV-2 representative of the most widespread strain in Europe during the first wave of the pandemic, whose gnomic data are available at EBI under study accession number PRJEB38101 [28]. Gendrot and collaborators employed the IHUMI-3 strain for which genomic data are not available, thus making it difficult to establish the degree of widespread of the virus and its comparison with other strains. Our SPR data indicated that doxycycline did not interact with relevant binding sites of S or ACE2 proteins, as instead suggested by an in silico study [15]. It cannot be excluded that it may affect, at least on the pseudotyped retroviral vector, the integrity of the virus lipidic envelope, suggested to be important for the virus integrity [32].

Although the conclusion of our study is somewhat disappointing, it is paradigmatic of an experimental approach aimed at developing an integrated multidisciplinary platform. 

To avoid wasting precious time and resources, we therefore believe that it is necessary to set very stringent experimental criteria in the preclinical phase, including in the platform infectious studies with SARS-CoV-2, before moving on to futile clinical trials.

This strategy may help develop a scientifically sound procedure for selecting potentially active molecules at the preclinical stage.

## Figures and Tables

**Figure 1 viruses-13-01745-f001:**
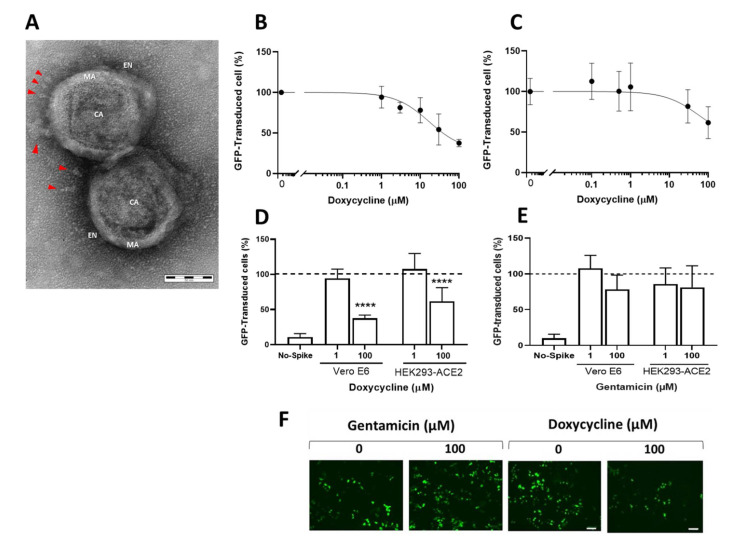
Doxycycline inhibited the transduction of pseudotyped retroviral vector exposing the SARS-CoV-2 S protein. (**A**) Representative image of two isolated pseudotyped retrovirus particles exposing the SARS-CoV-2 S protein. CA, capsid; EN, envelope; MA, matrix; spikes are indicated by red arrowheads. Scale bar, 50 nm. Dose-response effect of doxycycline in VeroE6 (**B**) and HEK293-ACE2 (**C**) cells. The *y*-axis shows the mean ± SD percentage of GFP-transduced cells in relation to control cells. The top limit was set as the average percentage for the vehicle-only control of this assay. Effects of 1 or 100 µM doxycycline (**D**) and 1 or 100 µM gentamicin (**E**) on transduction of the pseudotyped retroviral vector with SARS-CoV-2 S protein in Vero E6 and HEK293-ACE cells. Data are means ± SD of the percentage of GFP-transduced cells in relation to control cells transduced with vehicle only (dotted line). The percentage of Vero E6 cells transduced with the retroviral vector without SARS-CoV-2 S protein (No-Spike) is reported as negative control. **** *p* < 0.0001 vs. vehicle according to one-way ANOVA and Bonferroni’s post hoc test. (**F**) Representative fluorescence microscopy images of HEK293-ACE2 cells infected with the retroviral vector and treated or not with gentamicin or doxycycline. Scale bar, 100 µm.

**Figure 2 viruses-13-01745-f002:**
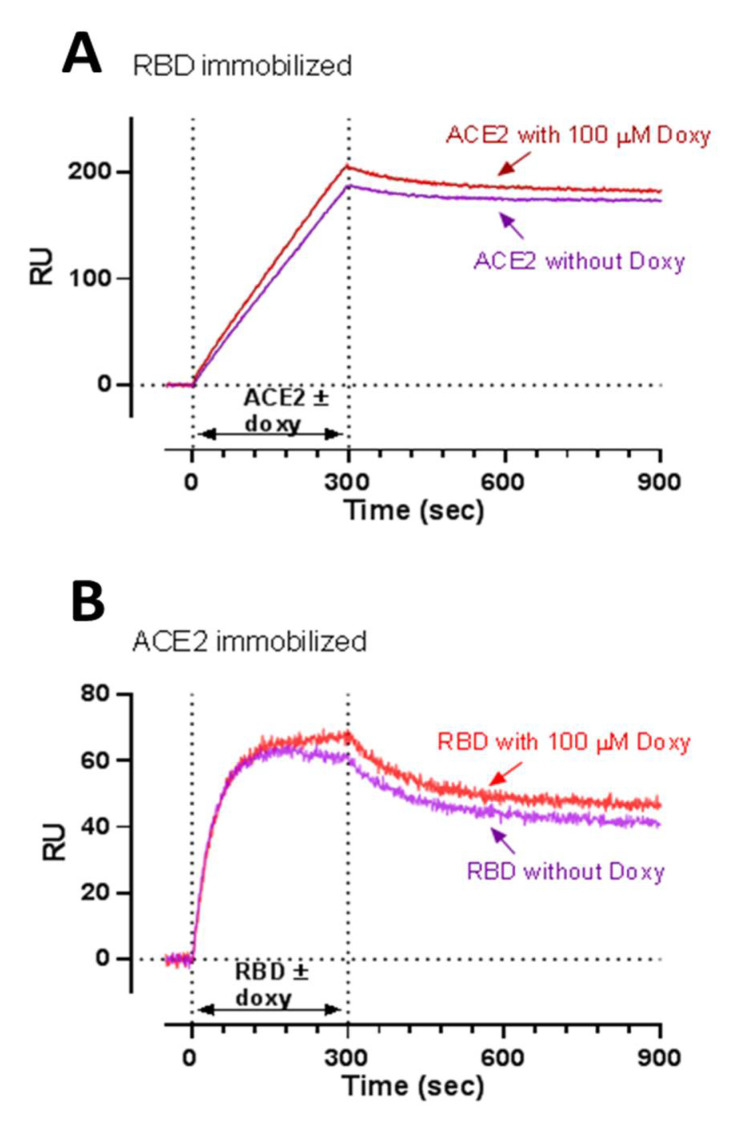
Surface plasmon resonance analysis showing no inhibitory effect of doxycycline on the ACE2-RBD interaction. ACE2-RBD interaction was evaluated either by flowing ACE2 (10 nM) over immobilized RBD (**A**) or, vice versa, RBD (60 nM) over immobilized ACE2 (**B**). This interaction was evaluated either in the absence or presence of 100 μM doxycycline. In particular, we preincubated the proteins for 60 min at room temperature with doxycycline or its vehicle, and then injected the mixture over the chip-immobilized protein binding partners for 300 s, as indicated. The graphs show the sensorgrams after subtraction of the SPR signal on reference surfaces (anti-Fc antibody for RBD; or anti-Flag antibody for ACE2). These are representative sensorgrams from one experimental session. Results were similar in three independent sessions.

**Figure 3 viruses-13-01745-f003:**
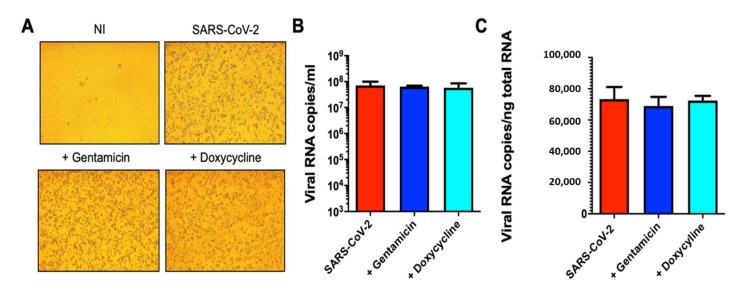
Doxycycline did not inhibit SARS-CoV-2 authentic virus replication. Vero E6 cells were pretreated for 4 h with doxycycline or gentamicin (100 μM), then infected with SARS-CoV-2 at a multiplicity of infection (MOI) of 0.01 for 1 h at 37 °C with doxycycline or gentamicin (100 μM). Cells were then washed and cultured for 48 h in fresh medium containing doxycycline or gentamicin (100 μM). Non-infected cells (NI) or cells infected without doxycycline or gentamicin treatment (SARS-CoV-2) were used as controls. (**A**) Cells were imaged with an optical microscope to detect typical SARS-CoV-2-induced cytolytic effects (original magnification 10×). (**B**) Viral yield was quantified in the cell supernatant by qRT-PCR. (**C**) Quantification of SARS-CoV-2 genomes at the intracellular level by qRT-PCR. Data are means ± SD of at least three independent replicates.

## Data Availability

Raw data can be required to the corresponding authors.

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
