# Peer review of "Doxycycline Inhibition of a Pseudotyped Virus Transduction Does Not Translate to Inhibition of SARS-CoV-2 Infectivity"

_viruses, 2021, doi:10.3390/v13091745_

Round 1

Reviewer 1 Report

Authors designed in vitro binding tests and infection studies with a pseudotyped virus expressing the spike protein and a clinically isolated SARS-CoV-2 strain. Results indicated that Doxycycline inhibited the transduction of the pseudotyped virus but did not affect the entry and replication of SARS-CoV-2.

This is a well written manuscript. Sections are clearly presented. The methods and results are presented in detail.

There are many figures in this manuscript. It might be helpful to remove some figures, and to keep those related to the main findings of your work.

Introduction

Page 2, last paragraph, third sentence: “The study was stopped for futility last March because…”

Please delete the word “last" and state clearly the month and year.

Author Response

Q1. There are many figures in this manuscript. It might be helpful to remove some figures, and to keep those related to the main findings of your work.

As requested, some figures have been removed and are now shown as Supporting Information. Figure 2 is now Figure S4 and Figure 3 is Figure S5.

Q2. Introduction, Page 2, last paragraph, third sentence: “The study was stopped for futility last March because…”Please delete the word “last" and state clearly the month and year.

The sentence has been modifies as follows:The study was stopped for futility in March 2021 because…..”

Reviewer 2 Report

Given the pleiotropic effects of doxycycline as an anti-viral agent, many clinical trials have started during the pandemic to determine the therapeutic efficacy of doxycycline alone or in combination for COVID-19. Although some the clinical observations are anecdotal and conflicting, in vitro characterization of the mechanisms underlying the possible effect of tetracyclines on SARS-CoV-2 are urgently needed. Toward this end, the authors investigated the activity of doxycycline against SARS-CoV-2 infection by developing in vitro binding assays and conducting infection studies with a pseudotyped virus expressing the spike protein, as well as a clinically isolated SARS-CoV-2 strain. Doxycycline effectively inhibited the transduction of pseudotyped virus but did not affect the entry and replication of SARS-CoV-2. Despite the negative results, the findings are important to disseminate and will contribute toward defining more stringent categories of judgment to improve the initial selection of potentially active molecules. The paper is well-written; experimental methods, design and results are clearly presented both in the text and figures. The discussion is thorough, and the conclusions are supported by the results. Although samples were collected from a human subject, there is no mention of protocol review by a human subjects review board or informed consent addressed.

Author Response

Q1. Although samples were collected from a human subject, there is no mention of protocol review by a human subjects review board or informed consent addressed.

Virus isolation is a laboratory practice usually performed in the hospital setting for diagnostic purposes and therefore does not require any informed consent or approval by an ethical committee.